# Exploring the Impact of Subtle Differences in the Chemical Structure of 1-Alkylsulfates and 1-Alkylsulfonates on Their Interactions with Human Serum Albumin

**DOI:** 10.3390/molecules29194598

**Published:** 2024-09-27

**Authors:** Ola Grabowska, Ankur Singh, Krzysztof Żamojć, Sergey A. Samsonov, Dariusz Wyrzykowski

**Affiliations:** Faculty of Chemistry, University of Gdańsk, Wita Stwosza 63, 80-308 Gdańsk, Poland; ola.grabowska@ug.edu.pl (O.G.); a.singh.706@studms.ug.edu.pl (A.S.); krzysztof.zamojc@ug.edu.pl (K.Ż.)

**Keywords:** 1-alkylsulfonates, 1-alkylsulfates, human serum albumin, protein–ligand interactions, isothermal titration calorimetry, fluorescence spectroscopy, molecular dynamics

## Abstract

The objective of this study was to examine the interactions between anionic surfactants, specifically 1-alkylsulfonates (KXS) and 1-alkylsulfates (SXS) ions, with human serum albumin (HSA). A combination of experimental techniques, including isothermal titration calorimetry (ITC), steady-state fluorescence spectroscopy (SF), and molecular dynamics-based approaches was employed to gain a comprehensive understanding of these processes. It has been demonstrated that the subtle variations in the charge distribution on the anionic surfactant headgroups have a significant impact on the number of binding sites, the stoichiometry of the resulting complexes, and the strength of the interactions between the surfactants and the protein. Additionally, we established that the affinity of the investigated ligands to specific regions on the protein surface is governed by both the charge of the surfactant headgroup and the length of the aliphatic hydrocarbon chain. In summary, the findings highlight the crucial role of charge distribution on surfactant functional groups in the binding mode and the thermodynamic stability of surfactant–protein complexes.

## 1. Introduction

The amphiphilic structure of 1-alkylsulfonates (KXS) and 1-alkylsulfates (SXS), where X represents the number of carbon atoms in the hydrocarbon chain, classifies them as surface-active compounds (surfactants) with the capacity to reduce surface tension between liquids or between a liquid and a solid surface. The remarkable properties of 1-alkylsulfates and the related 1-alkylsulfonates make them incredibly versatile and applicable across numerous industrial and technological fields. These applications include the detergent industry, food chemistry, drug delivery, cosmetic preparation, and the life sciences [1,2,3,4,5].

1-Alkylsulfonates are characterized by the presence of a sulfonate group (-SO_3_), whereas 1-alkylsulfates, closely related to their sulfonate counterparts, boast a sulfate group (-OSO_3_) in their molecular structure. In contrast to SXS, sulfonates feature a direct linkage between the sulfur atom of a headgroup and a carbon chain of specific lengths (Figure 1).

These seemingly subtle differences in the structure of SXS and KXS have a significant impact on the overall charge of the headgroup of the sulfate and sulfonate ions [6]. This, in turn, is reflected in the processes determined by (de)hydration effects [7,8]. These differences, in turn, have far-reaching implications for the physicochemical properties of SXS and KXS, including critical micelle concentration (CMC), critical aggregation concentration (CAC), micellar micropolarity, and aggregation number.

One noticeable distinction lies in the critical micelle concentrations (CMCs) of specific examples like sodium dodecyl sulfate (S12S) and sodium 1-dodecanesulfonate (K12S). For S12S, the CMC is 8.3 mM, whereas for K12S, it is 10.8 mM [9]. Significant differences are exhibited in the Krafft points of KXS, which are noticeably higher than those of SXS (the Krafft points of S12S and K12S are 282.15 K and 311.15 K, respectively) [10]. This disparity in Krafft points emphasizes the distinct thermal behaviour and solubility characteristics of 1-alkylsulfonates and 1-alkylsulfates. Interestingly, we have recently proven, in one specific case, that K12S molecules are not recognized by S12S as the ones with a different structure and consequently are allowed to participate in the formation of the S12S micelles [11]. However, it can be postulated that in this specific case, K12S participates in the formation of S12S micelles due to the hydrophobic chain length being identical to that of S12S, and the differences in charge distribution on the headgroups being of less importance.

In the realm of surfactant–protein interactions, extensive research has been conducted on classical surfactants [12,13,14,15,16,17,18,19]. These studies have primarily focused on factors such as the surfactant topology (monomeric or micellar), chemical structure (cationic, anionic, nonionic, hydrocarbon chain length), and experimental conditions (pH, temperature) [12,20,21]. However, relatively little attention has been paid to the distinct headgroup structures of anionic surfactant components [6]. These headgroup structures have the potential to significantly influence ligand–protein interactions. Thus, the objective of this study is to delve into the impact of the differing headgroup charges between KXS and SXS (Figure 1) on their interactions with human serum albumin (HSA). By exploring this aspect, we aim to shed light on the intricate interplay between surfactant headgroup charge and protein interactions.

## 2. Results and Discussion

### 2.1. Interactions of 1-Alkylsulfonates (KXS) and 1-Alkylsulfates (SXS) Ions with Human Serum Albumin (HSA)

The binding of KXS and SXS ions to human serum albumin (HSA) has been analysed using isothermal titration calorimetry (ITC) and fluorescence spectroscopy. These methods offer essential insights into crucial information regarding ligand–protein complexes. This includes determining the stoichiometry of the investigated complexes, the number of binding sites on the protein surface and the strength of the interactions (known as binding affinity). In addition, thermodynamic parameters such as enthalpy and entropy changes due to the established interactions provide valuable information that contributes to a deeper understanding of the molecular interactions under study (Table 1).

#### 2.1.1. The Stoichiometry and the Number of the Binding Sites

The relatively minor structural differences between KXS and SXS have a significant impact on their binding interactions with human serum albumin (HSA). These differences are manifested in the stoichiometry of the ligand–protein complexes and the number of binding sites. Conditional parameters for the interactions under study are summarized in Table 1, whereas representative binding isotherms are shown in Figure 2. A significant difference in the interactions of KXS and SXS with the protein is reflected in the number of binding sites. In the case of the K12S and K10S ligands, HSA was found to have only one binding site, leading to the formation of complexes with a KXS:HSA molar ratio of approximately 1:1. In contrast to KXS, the S12S and S10S ligands bind to two separate sites on the HSA protein. The first site accommodates one mole of SXS per one mole of the protein, while the second site binds four moles of SXS (Figure 1).

#### 2.1.2. The Thermodynamic Parameters of the Interactions

In general, the interaction between surfactants and proteins is primarily driven by hydrophobic interactions between the ligands and the hydrophobic regions of the protein. In these cases, the stability of the resulting complexes is influenced by the length of the hydrophobic ligand; specifically, the stability of the ligand–protein complex tends to increase with a greater number of carbon atoms in the hydrocarbon chain. However, for the ligands under study, the charged headgroups of KXS and SXS can also participate in electrostatic interactions with the positively charged regions of HSA, influencing binding characteristics and conformational changes in the protein.

The log*K* and Δ*H* values indicating the affinity of K12S and K10S for human serum albumin are approximately equal (Table 1). A comparable situation in binding interactions also applies to S12S and S10S. This indicates that the difference in hydrocarbon chain length (C12 vs. C10) is insufficient to impact the binding strength of these ligands (Table 1).

It is noteworthy that, for all the systems studied, a negative value of Δ*H* indicates a significant contribution of electrostatic interactions, hydrogen bonds and/or Van der Waals interactions to the stabilization of the resulting ligand–protein complexes (Table 1).

The enthalpy–entropy compensation phenomenon is observed in the interactions studied. It is more pronounced for KXS-HSA interactions (Table 1). The more negative the value of Δ*H*, the lower the mobility of the interacting molecules, which is reflected in the low value of the entropy factor (Table 1). Furthermore, ab initio calculations showed that the α-methylene group of KXS had a partial negative charge, whereas the α-methylene group of SXS had a partial positive charge. Thus, the headgroup of KXS had a much smaller negative charge than SXS (for details see Section 2.3). Consequently, differences in the entropy change upon binding the ligands to HSA may be due to differences in the charge distribution of the headgroups of KXS and SXS.

#### 2.1.3. Effect of KXS and SXS on the Tryptophan Environment of Human Serum Albumin

To understand the effect of the studied compounds on the tryptophan environment of human serum albumin, the intrinsic fluorescence of the protein in the presence of increasing concentrations of K12S, K10S, S12S and S10S has been studied. HSA in 10 mM Caco buffer of pH 7 after excitation at 280 nm exhibits a single emission peak with a maximum at 343 nm (due to a tryptophan residue). On the addition of the ligands under study an enhancement in the protein fluorescence intensity can be observed (contrary to bovine serum albumin, where fluorescence quenching by anionic surfactants occurs [21,22,23]), which is consistent with the existence of a unique buried tryptophan residue in HSA with considerable static quenching in the native state [24]. Upon the binding, a removal of quenching occurs as a result of the interaction between the indole side chain of W214 residue with neighbouring quenching groups [22]. A very clear shift (up to 15 nm) of the band towards shorter wavelengths (Figure 3) additionally suggests that all selected for the studies compounds interact with HSA and induce a change of the microenvironment around the protein, with a tryptophan residue relocated to a more hydrophobic environment. The observed effect of the studied compounds on the fluorescence emission of HSA is similar to the interactions of the protein with—among others—syringin [25], salbutamol [26], and abiraterone [27]. It has been demonstrated that SXS has a more pronounced impact on emissions than KXS. Furthermore, the magnitude of the observed change is directly proportional to the length of the hydrocarbon chain, which is consistent with previous reports [28,29].

### 2.2. Potential Binding Sites for KXS and SXS on Human Serum Albumin

Different binding mechanisms determine the affinity of small molecules for HSA. Depending on the structural features of a ligand, several distinct regions with different binding mechanisms (involving hydrophobic interactions or polar forces) are available on the surface of albumin. To explore the potential binding sites of the studied ligands on the surface of the proteins and to gain insight into the differences in the binding of KXS and SXS, molecular dynamics simulations as well as ITC competitive experiments were performed.

#### 2.2.1. A Molecular Dynamics Simulation Approach to Binding Site Determination

The computational simulation hypothesis was based on experimental data indicating the existence of multiple binding sites on human serum albumin (HSA), as well as observable differences in the binding affinity of KXS and SXS ligands to this protein. To further investigate the binding sites and ligand affinities on HSA, molecular dynamics simulations were performed using the AMBER software package and LIE (Linear Interaction Energy) methodology was used for binding free energy calculations.

The simulations confirmed the presence of multiple binding sites on human serum albumin (HSA) for the four ligands investigated: S12S, S10S, K12S, and K10S (Figure 4). These findings align with experimental data, which also indicated the existence of distinct binding sites for these ligands on the protein.

Ligands S10S and S12S revealed the ability to bind at two distinct sites on HSA, specifically within the IIIA, IIA, and IB domains. Free energy calculations showed that the binding affinity for S10S localized in the proximity of 5 Å from the analyzed putative binding sites was −31.1 ± 3.9 kcal mol^−1^ at the first binding site, designated as “Site 1,” and −32.3 ± 4.3 kcal mol^−1^ at the second binding site, referred to as “Site 2” (Figure 4). For S12S, the binding affinity was also slightly higher in “Site 2”, −34.8 ± 4.2 kcal mol^−1^ compared with “Site 1” (−32.1 ± 5.0 kcal mol^−1^). In contrast, ligands K10S and K12S were found to exhibit only one binding site, which was located in the IIIA domain of HSA.

The free energy analysis indicated that the IIIA domain has a high likelihood of serving as the binding site for ligands K10S and K12S. The binding affinities of K10S and K12S to HSA were found to be comparable, irrespective of the length of the hydrophobic moiety of the ligand, supporting our hypothesis that chain length does not significantly affect binding affinity in this instance. Notably, ligands S10S and S12S displayed a shared second binding site, which is situated close to the binding site for K10S and K12S within the IIIA domain. These findings suggest the presence of a shared binding site in this region for the ligands S10S, S12S, K10S, and K12S. However, the first binding sites for S10S and S12S were distinct from one another. For S12S, the binding site is located near the interface of the IB and IIIA domains, while for S10S, it is situated near the upper part of the IB domain. The favourable binding energy at the second binding site for S10S and S12S, along with the first binding site for K10S and K12S suggested potential interactions between these ligands and human serum albumin (HSA) in these predicted binding sites.

#### 2.2.2. Competitive ITC Displacement Assays for Binding Site Determination

To verify the MD simulation findings, the ITC competitive experiments were conducted. In this approach, the S12S molecules have been employed as the HSA site markers, namely “Site 1” (located in the HSA cavity between the IIA and IB domains) and “Site 2” (located in the IIIA domain of HSA). Saturation of the HSA protein with the S12S ligand may hinder or even prevent the binding of other anions (i.e., S10S, K12S, K10S), provided that they compete with S12S for the same binding site in HSA.

ITC data revealed that HSA saturated with S12S at a molar ratio of S12S:HSA = 1.5:1 is still capable of binding K12S and K10S ligands (Figure 5, Table 2). In the case of S10S, its interactions with HSA are hindered. This is manifested by the presence of only one binding site for S10S (“Site 2” located in subdomain IIIA), and a reduced number of moles of S10S (ca. 2) bind to one mole of HSA under these specific conditions (Table 2).

A different scenario is observed for HSA with fully saturated binding sites by S12S (S12S:HSA molar ratio = 6:1). In such a case, there is a lack of evidence for the interactions of HSA with K12S and K10S (Figure 6, Table 2). Thus, according to the MD simulation results, it can be expected that the potential binding sites of KXS will be located in subdomain IIIA. 

It is noteworthy that despite the complete saturation of HSA with S12S molecules, S10S molecules, unlike 1-alkylsulfonates, are still able to interact with HSA (Figure 6, Table 2). However, compared to the interaction of S10S with free HSA, there is only one S10S binding site on the HSA surface under such specific conditions. This observation is consistent with theoretical results indicating that S10S and S12S have an affinity for one region of the protein (in the IIIA domain), while the other binding region of these ligands is different. For these reasons, complete saturation of HSA with the S12S ligand still allows S10S to bind near the interface of domains IB and IIIA.

### 2.3. The Hydration Patterns of 1-Alkylsulfonates (KXS) and 1-Alkylsulfates (SXS) Ions

To gain insight into the differences in physicochemical properties between KXS and SXS, the spatial distribution of water oxygen atoms in relation to the sulfate group was investigated using the radial distribution function (RDF). The obtained data indicate that significantly different affinities of KXS and SXS for HSA may be influenced by differences in the hydration patterns of the polar headgroups of the ligands (Figure 7). The observed differences in hydration properties originated from the charge distributions in the headgroups of SXS and KXS that are essentially different (Figure 8). While the charge differences of the atoms in SO_3_^-^ are negligible, the main differences in the charge distribution of the headgroup correspond to the negatively charged oxygen atom present in SXS and absent in KXS. The presence of this oxygen atom strongly affects the charge of the adjacent carbon atom and hydrogen atoms bound to it. This effect is already very weak for the next carbon atom in the chain. The presence of an additional oxygen atom in SXS in comparison to KXS, therefore, leads to a more polar nature of the SXS headgroup. This, in turn, enhances the strength of the electrostatic interactions that this group can establish with its environment, affecting the organization of water molecules in its hydration shell. As a result, it can potentially influence both the enthalpy of binding to a protein and the entropy of dehydration when a direct contact with the protein is maintained.

## 3. Materials and Methods

### 3.1. Materials

Human serum albumin (HSA, ≥96%), sodium dodecyl sulfate (S12S, ≥99.0%), sodium n-decyl sulfate (S10S, 98%), sodium 1-dodecanesulfonate (K12S, ≥99%), sodium 1-decanesulfonate (K10S, ≥99%), sodium cacodylate trihydrate (Caco, ≥99%) were obtained from Merck (Warsaw, Poland) and employed as received without further purification. All solutions were prepared in the 10 mM Caco buffer (pH 7).

### 3.2. Methods 

#### 3.2.1. Isothermal Titration Calorimetry (ITC)

Experiments were conducted using AutoITC isothermal titration calorimeter (MicroCal Inc. GE Healthcare, Northampton, MA, USA). A 1 mM solution of SXS or KXS (where X = 12, 10) was injected into the reaction cell, which contained: (a) a 0.04 mM solution of free HSA, (b) a mixture of HSA:S12S in molar ratios of 1:1.5 and 1:6, respectively. ITC experiments were conducted under specific conditions: temperature 298.15 K, cell volume 1.4491 mL, injection volume 10.02 μL (29 injections, 2 μL for the first injection only), injection interval 240 s, stirring speed 300 r.p.m., reference power 10 μcal s^−1^. The thermodynamic parameters (*K*_ITC_ and Δ*H*_ITC_) were obtained directly from ITC measurements by fitting isotherms (using nonlinear least-squares procedures) to a model that assumes a single set and two sets of binding sites, respectively for 1-alkylsulfonates (KXS) and 1-alkylsulfates (SXS). 

#### 3.2.2. Steady-State Fluorescence Spectroscopy (SF)

The concentration of the human serum albumin was measured spectrophotometrically using an extinction coefficient equal to $\varepsilon_{280}^{HSA}=32930\;M^{-1}{cm}^{-1}$. For this purpose, the UV absorption spectra were recorded on a Perkin Elmer Lambda 650 (Waltham, MA, USA) UV/Vis spectrophotometer. Fluorescence emission spectra were recorded on a Cary Eclipse Varian (Agilent, Santa Clara, CA, USA) spectrofluorometer equipped with a 1.0 cm multi-cell holder and temperature controller. Experiments were performed in 10 mM Caco buffer (pH 7). Excitation and emission slit widths were fixed at 5 nm. The excitation wavelength was always set at 280 nm. Fluorescence emission spectra of HSA (2.2 µM) were recorded in the range 290–500 nm in the absence and presence of S10S, K10S, S12S and K12S while increasing the concentration of each up to 12.3 µM. In the fluorescence titration experiments, 2 mL of HSA was titrated with 5 µL of S10S, K10S, S12S, and K12S (1 mM) stock solutions added in five consecutive additions. All experiments were performed at 298.15 K.

#### 3.2.3. Molecular Dynamics (MD) Simulations

The structure of HSA was obtained from the PDB (PDB ID: 5 × 52, 3.00 Å). The structures of SXS and KXS were built in xleap module of AMBER 20 package [30]. The corresponding libraries for AMBER package have been created with Antechamber using a standard procedure for GAFF force field [31] and HF/6-31G* RESP charge derivation methodology [32]. Fifteen ligand molecules of each type for each simulation were placed randomly around HSA. GAFF and ff14SB [33] force field parameters were used for the ligands and the protein, respectively. These molecular systems were solvated in a truncated octahedron TIP3P water periodic box with a distance of 15 Å from the solute to the box and neutralized by counter-ions. MD simulations were preceded by two steps of energy-minimization: 1500 cycles of steepest descent and 1000 cycles of the conjugate gradient with harmonic force restraints of 10 kcal*(mol·Å)^−1^ on protein and ligand atoms, then 3000 cycles of steepest descent and 3000 cycles of conjugate gradient without restraints. This was followed by heating of the system from 0 to 300 K for 10 ps, and a 100 ps MD equilibration run at 300 K and 105 Pa in NPT. After this equilibration procedure, 100 ns of productive MD runs were performed in periodic boundary conditions in NPT with an 8 Å cut-off for non-bonded interactions and the particle mesh Ewald method applied to treat long-range electrostatic interactions. Energetic post-processing of the trajectories was done using linear interaction energy (LIE) with a dielectric constant of 80. The obtained structures were visualized in VMD [34].

## 4. Conclusions

The objective of this study was to investigate the interactions between 1-alkylsulfonates (KXS) and 1-alkylsulfates (SXS) ions with human serum albumin (HSA). To gain insight into the binding characteristics of these two types of surfactants, a comparative analysis was conducted.

The results revealed that even small differences in the charge distribution on the anionic surfactant headgroups have a significant impact on the interactions between the surfactants and the protein. This influence is evident in the number of binding sites and the stoichiometry of the resulting complexes, as well as the strength of the interactions.

Furthermore, both single and shared regions were identified on the protein surface where KXS and SXS bind. The affinity of individual ligands for these sites depended on the surfactant headgroup’s charge and the hydrocarbon chain length (X = 12 or 10). Understanding these binding regions provides insight into the specific interactions between surfactants and proteins.

It is important to note that the parameters obtained for these interactions depend on the pH of the system, the type of buffer used, and its concentration (ionic strength). Therefore, they can only be compared with parameters determined under identical experimental conditions. Nevertheless, the data obtained under the same experimental conditions confirmed previous findings regarding the significant impact of the hydrophilic moieties of KXS and SXS on their physicochemical properties. By gaining a deeper understanding of these subtle differences, we can improve our interpretation of processes involving 1-alkylsulfates and 1-alkylsulfonates, particularly in the context of protein–ligand interactions.

In conclusion, our study highlights the importance of charge distribution on surfactant functional groups and hydrophobic chain lengths in determining the binding characteristics and thermodynamic stability of surfactant–protein complexes.

## Data Availability

The data presented in this study are available on request from the corresponding authors.

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
