# Peer review of "Exploring the Impact of Subtle Differences in the Chemical Structure of 1-Alkylsulfates and 1-Alkylsulfonates on Their Interactions with Human Serum Albumin"

_molecules, 2024, doi:10.3390/molecules29194598_

Round 1

Reviewer 1 Report

Comments and Suggestions for Authors

In this study, the authors used various biophysical experiments including ITC, SF and CD to investigate the interactions between four different species of surfactants and HSA. The data was clearly presented, and results were beneficial for many different fields. However, I still have a few questions about the experimental data:

1.  By observating the figures of ITC data, only S12S showed a clear two-stage binding pattern, and S10S will probably pass using a one-site binding model. Could the authors provide raw data so readers and evaluate by ourselves to see there was no overfitting?

2. It is well known that albumin has multiple fatty acid binding sites. However, in the ITC analysis, different binding sites were combined as one site that binds multiple surfactants. Is it possible to further distinguish the nuance differences of these different sites?

3. Can the author use alternative experiments to confirm the binding stoichiometry between HSA and these surfactants? For example, LC-mass analysis or SPR.

4. Since the authors have tested fluorescence emission of tryptophan in HSA, will they also try thermoshift (DSF) assay to check how the surfactants are affecting stability of HSA? The data could be used as a compliment as ITC data.

Reviewer 2 Report

Comments and Suggestions for Authors

Grabowska et. al, have conducted the study of the binding of 1-alkylsulfonates (KXS) and 1-alkylsulfates (SXS) ions with human serum albumin 11 (HSA). And interestingly, they found the subtle structure difference between head groups and chain lengths of SXS and KXS can have a great influence on their binding to the HSA. The study is clearly written and interesting to read. I recommend the publication of this study. Only some minor revision may be needed.

1. A clear difference of TΔS values of SXS and KXS of the ITC result in Table 1 can be observed. The entropy terms are critical to the overall binding affinities. The authors are encouraged to discuss more about the potential correlation of entropy change upon binding to HSA with the structure of SXS and KXS surfactant.

2. The MD simulation results are really interesting. The important residues that may contribute to the interaction with KXS or SXS can be studied by calculating the overall contacting frequency of each residues to surfactant molecules during the MD simulations. And maybe different important residues contributing to the interaction with surfactant molecules in the HSA/SXS and HSA/KXS systems can be observed.

3. As to Fig. 8, the RDF results of oxygen atoms relative to the KXS or SXS are presented, which shows relatively not so rich information. The authors are suggested to carry out more detailed analyses of relative orientation and clustering of OH of water molecules to the oxygen atoms of surfactant head groups.

4. And the partial charges of KXS and SXS molecules, especially head groups, are suggested to present in a figure for a clear comparison of the charge distributions of the head groups. 

5. Besides, the authors are encouraged to discuss more about the correlation of hydration patterns of different surfactant head groups with the binding of surfactant to specific binding sites of HSA (maybe hydrophobic pockets).

Reviewer 3 Report

Comments and Suggestions for Authors

1. The results section presents the experimental data, but the interpretation of the figures is overly lengthy and fails to highlight the key findings clearly. The complexity of data presentation, especially regarding binding sites and thermodynamic parameters, may confuse readers. Simplifying the presentation or adding annotations to the figures is recommended for better comprehension.

Although the discussion section introduces some interesting hypotheses, the overall argument is weak and does not sufficiently explain the mechanisms behind the experimental results. For example, the discussion on the impact of KXS and SXS chain lengths on protein thermal stability is superficial and lacks in-depth analysis and comparison with existing literature.

2. The experimental methods are described in detail, but some crucial details are missing. For example, the molecular dynamics simulation methods are not fully explained in terms of specific parameters and conditions. Additionally, the rationale behind the choice of these methods is not adequately justified, leaving readers to question why these particular techniques were chosen to study the interactions between KXS, SXS, and human serum albumin (HSA).

3. The conclusion section generally summarizes the findings but raises doubts about the generality and applicability of the conclusions. The wording is too broad, failing to clearly articulate the novelty and practical implications of the study. Furthermore, the limitations of the study and directions for future research should be discussed more thoroughly.

4. The language is accurate, but there are some grammatical and spelling errors that affect readability. In terms of formatting, the numbering and citation of some figures are inconsistent, and it is recommended to strictly follow the journal's submission guidelines.

Specifically, the text formatting of the annotations for many of your images and tables is quite strange, and many images that should be arranged vertically are instead arranged horizontally.

Reviewer 4 Report

Comments and Suggestions for Authors

The manuscript ”Exploring the Impact of Subtle Differences in the Chemical Structure of 1-Alkylsulfates and 1-Alkylsulfonates on their Interactions with Human Serum Albumin” by Ola Grabowska et al. aims to explore how slight structural differences between 1-alkylsulfates (SXS) and 1-alkylsulfonates (KXS) affect their interactions with human serum albumin (HSA). The study utilizes a combination of experimental techniques such as isothermal titration calorimetry (ITC), fluorescence spectroscopy, and circular dichroism (CD), along with molecular dynamics simulations, to understand these interactions. While the subject is interesting and the findings could have implications in fields like drug delivery and protein-surfactant interactions, the manuscript has several minor critical issues that may be addressed.

Major Concerns:

Reliance on Computational Models: Similar to other works in the field, this manuscript heavily relies on molecular dynamics simulations to “confirm” experimental data.  

“The simulations confirmed the presence of multiple binding sites ….” . Experimental data  confirms  the  hypothesized MD binding sites. 

Experimental validation of these predictions, perhaps through techniques like X-ray crystallography or NMR, would significantly strengthen the conclusions. The study would also benefit from techniques that can empirically map binding sites, such as mutagenesis or site-directed labeling, to verify the proposed interaction sites on HSA.

The manuscript provides thermodynamic parameters from ITC, yet the discussion of these results is somewhat ambiguous. The authors claim that both hydrophobic and electrostatic interactions play a role, but the precise nature of these interactions is not well-explained. For instance, how exactly do small changes in the headgroup charge influence the interaction energy? Furthermore, the role of the hydrophobic chain length, while mentioned, is not systematically explored beyond a simple comparison of 10 to 12. 

Probably addition of  8, 9, 11 etc. chain lengths will  improve the details of the  analysis on how chain length correlates with binding affinity.

The discussion surrounding the surfactant-protein interactions is somewhat general and lacks mechanistic depth. The authors mention that SXS and KXS interact with HSA through hydrophobic and electrostatic forces, but they do not provide detailed structural insights or discussion on how these forces compete or cooperate. A more nuanced analysis, perhaps through complementary structural studies, could make this section more impactful.

Minor Concerns:

 While the study reports standard deviations for ITC-derived thermodynamic parameters, the manuscript does not discuss the reproducibility of the fluorescence and CD spectroscopy results. Given the complexity of surfactant-protein systems, small changes in experimental conditions could significantly affect the results. Including a brief discussion on experimental reproducibility would strengthen the study’s rigor.

The discussion of the thermodynamic parameters in Table 1 lacks sufficient explanation. Furthermore, terms like “conditional thermodynamic parameters” and “binding isotherms” should be clarified for non-specialist readers. 

Some recent references may be added when discussing key aspects of the work. For example, the manuscript refers to previous studies on surfactant-protein interactions but recent literature might provide alternative interpretations or comparisons. 

Round 2

Reviewer 3 Report

Comments and Suggestions for Authors

The manuscript has been significantly improved, and I agree with the positive review. It is ready for acceptance.